# Weather Research and Forecasting—Fire Simulated Burned Area and Propagation Direction Sensitivity to Initiation Point Location and Time

Amy DeCastro [1,*,†], Amanda Siems-Anderson [1], Ebone Smith [2], Jason C. Knievel [1], Branko Kosović [1], Barbara G. Brown [1] and Jennifer K. Balch [3]

1   National Center for Atmospheric Research, Research Applications Laboratory, Boulder, CO 80305, USA; aander@ucar.edu (A.S.-A.); knievel@ucar.edu (J.C.K.); branko@ucar.edu (B.K.); bgb@ucar.edu (B.G.B.)
2   Virginia Tech, Blacksburg, VA 24061, USA; esmith97@vt.edu
3   Earth Lab, University of Colorado Boulder, Boulder, CO 80303, USA; jennifer.balch@colorado.edu
*   Correspondence: decastro@ucar.edu
†   Current address: 3450 Mitchell Ln, Boulder, CO 80301, USA.

**Abstract:** Wildland fire behavior models are often initiated using the detection information listed in incident reports. This information carries an unknown amount of uncertainty, though it is often the most readily available ignition data. To determine the extent to which the use of detection information affects wildland fire forecasts, this research examines the range of burned area values and propagation directions resulting from different initiation point locations and times. We examined the forecasts for ten Colorado 2018 wildland fire case studies, each initiated from a set of 17 different point locations, and three different starting times (a total of 520 case study simulations). The results show that the range of forecast burned area and propagation direction values is strongly affected by the location of the initiation location, and to a lesser degree by the time of initiation.

**Keywords:** wildland fire detection; wildland fire behavior model; ignition point; sensitivity study

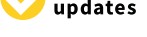



## 1. Introduction

Wildland fire is an important, regenerative process in many ecosystems globally. For example, ponderosa pine forests in the Western United States (U.S.), Mediterranean pine stands in Italy, and grass-land systems in South Africa experience regeneration post-wildfire [1–3]. However, in many regions, climate change and human activity have increased wildfire activity, resulting in uncontrollable fires that have endangered life and property [4,5]. This trend is likely to continue. Climate change has shifted fuel characteristics by changing the suitability of the environment and the mechanisms that determine the rise and fall of species populations [6,7]. In tropical climates, climate-induced droughts decrease photosynthesis, and increase tree mortality and autotrophic respiration (a large source of $CO_2$), promoting wildfire by providing available fuel in previously fuel-limited systems [7]. Human-related ignitions have expanded the extent of wildfire by bringing ignitions into regions with higher fuel moisture and net primary productivity. In the U.S. fire occurrence is increasing in areas with historically low lightning strike density (Mediterranean California climate) and in areas of low lightning occurrence and dry fuels (Eastern Temperate Forest climate) [8]. Flammable landscapes with intermediate population densities in the Western U.S. and Southeastern Australia are particularly affected by extreme wildfire events. However, the more densely populated northern Mediterranean Basin with a similar climate is less prone. This difference is thought to be due to regional land use, which reduces available fuels [9]. In the U.S., indigenous and traditional fire management systems (similar to the effects seen in the Mediterranean Basin) have been replaced by operational, large-scale firefighting and fuel management practices. These practices have

also played a role in increasing the number of fires [10], and have contributed to the risk of more severe wildfires [11–13].

The increase in wildfire frequency and severity has motivated discussion on mitigation and prevention. Current management strategies, which focus heavily on suppression are not enough given the increasing wildland-urban interface (WUI) and climate conditions ripe for wildfire [14]. Response to active fire is aided by wildfire propagation forecasts generated through operational-use wildfire rate of spread models [15]. These models help anticipate when and where a wildfire is likely to propagate given a set of inputs describing weather, terrain, and fuel conditions. Significant effort has been made to develop accurate forecasting capabilities; however, wildfire behavior models continue to have uncertainties due to input data and changing fire regimes that lessen their utility. In a 2013 study of 47 different fire behavior models, the mean percent error of fire rate of spread fell between 20–310% [16]. A later study found that while mean errors have decreased overall for fire prediction models, models for operational application have not shown significant improvement [17]. Persistent uncertainties still originate from input data sets, particularly wind speed, wind direction, fuel model assignment, and ignition location and timing data [18].

Wildland fire behavior models are often initiated using an estimated ignition point (referred to here as a 'detection point') with an unknown amount of uncertainty in its time and location. Wildland fires may be detected through several sources, including volunteer reporting and optical smoke detectors, and are infrequently made at the very start of a fire with instrumentation that allows for accurate geolocation [19]. Actual ignitions (where and when the fire actually started) may never be known with absolute accuracy, and detection points are frequently used to initiate model simulations in their absence. The primary purpose of a detection point, at least initially, is to aid firefighters in locating a fire after it has been detected. The detection is a best guess at where and when a fire started and may be adjusted in the fire report as the fire progresses and more information is gained. For example, the point of origin for the Ryan Fire in Colorado, 2018, was estimated at 39.06757 latitude, −105.1443 longitude. Later in the fire event, that location estimate was shifted to 39.05989 latitude, −105.1439 longitude (a difference of about 855 m) [20]. As the precise time of the ignition is rarely known with certainty, the time of detection is recorded in the fire report with an estimate of the fire's size at the time of discovery. Along with the size estimate and time of discovery, the available fuels, topography, and atmospheric conditions help to give a sense of how long the fire has been burning. While this means that both the location and time of ignition are estimates with an unknown amount of uncertainty, they meet the needs of fire management. That is, the detection does not have to accurately represent the ignition time and location for effective wildland fire response, it simply needs to give a sense of the proximity and size of the fire.

However, in the context of wildland fire behavior modeling, the use of a wildland fire detection will have implications for the initialization parameters of the model. The simulated fire starts and propagates from the initial location using the fuel characteristics, wind forecast, and terrain features, associated with that point and the input time. As the fire detections are estimates, associations with these drivers may or may not be reflective of reality. Fuel characteristics and terrain features are heterogeneous in space, and wind vectors are specific to location and time, meaning that the initiation timing and location will affect the model output.

Furthermore, wildland fire behavior models are used to enhance our understanding of fire behavior through experimentation and comparison with observation data [21]. Both contexts, wildland fire research and operational use, benefit from an improved understanding of wildland fire behavior model limitations, including those introduced through input data. Sensitivity studies have investigated wildland fire behavior model sensitivity to fuel characteristics [22,23], the combination of meteorological conditions and fuels [24], and meteorological conditions [25]. Bachmann and Allgöwer's (2002) [26] study of uncertainty propagation within Rothermel's rate of spread equations comprehensively investigates

17 different input variables. Benali et al. (2017) [18] examined the uncertainty associated with fuels, winds, and ignition locations.

However, to the best of our knowledge, the sensitivity of wildland fire behavior models to initiation location and time is yet to be fully explored. How does the time and location of a wildfire detection point affect wildland fire behavior model forecasts? This research examines the range of burned area values and propagation directions resulting from different initiation point locations and times. We examined the forecasts for ten Colorado 2018 wildland fire case studies, each initiated from a set of 17 different point locations, and three different starting times. Our results show that the location of the initiation point is important to the simulation results, more so than the time of initiation. A brief investigation of the range of fuel, wind, and terrain inputs shows that the number of fuel models, wind speeds, wind directions, and terrain slopes and aspects are important predictors of the range of forecast burned area and propagation direction values. These results align with studies that have shown that environmental heterogeneity and compounding uncertainty affects wildland fire behavior model forecasts [18,26]. The methods of this study, including case study selection and model setup are given in the next section. A discussion of the results, conclusion, and recommendations follow.

These models are also used for operational management where fire forecasts aid in determining suppression strategies and the allocation of resources [15]. However, the uncertainty associated with wildland fire behavior model output can lead to diminished faith and use of fire forecasts [27]. Operational-use wildfire behavior models are used to estimate fire characteristics important to decision-support, such as the rate of spread, location, and direction of propagation. Accurately predicting these characteristics of fire behavior is critical to informing fire-response practices that will be most effective at keeping fire-fighters and nearby residents safe, determining appropriate mitigation strategies, and using suppression resources in efficient ways [15].

## 2. Materials and Methods

### 2.1. WRF-Fire

WRF-Fire [28,29] integrates the Weather Research and Forecasting (WRF) model [30] with a wildland surface fire-behavior physics module [31]. WRF is a community model used to study and forecast land-atmosphere interactions at different scales. Coupled with the fire-behavior physics module that implements the Rothermel rate of spread equations [32], WRF-Fire forecasts fire-atmosphere interactions driven by topographic, fuels, and weather data. The chosen model physics and dynamics options follow the operational WRF-Fire modeling setup, the Colorado Fire Prediction System, which was developed by the National Center for Atmospheric Research (NCAR).

The simulations for this study are performed using version 4.0.1 of WRF-Fire [33]. The simulations are run in WRF's large eddy simulation (LES) [34] configuration following Jimenez et al. (2018) [35], which allows for the fire to be initiated in a space in which the boundary layer turbulence is fully developed and generated through interactions with the land surface specific to the fire domain. The initial and boundary conditions are provided by High-Resolution Rapid Refresh (HRRR) [36] forecasts from the National Oceanic and Atmospheric Administration (NOAA) at 3 km resolution, downscaled to 1 km over 117 km by 117 km domain centered on the model initiation point. The extent of the model domain is approximately 13 km by 13 km, with two nested model domains at grid spacings of 111 m and 27.75 m. The aspect, elevation, and slope inputs come from the LANDFIRE topography products, a layer with 30 m spatial resolution [37,38]. The fuel models are brought into WRF-Fire through LANDFIRE's 40 Scott and Burgan Fire Behavior Fuel Models product, also at a 30 m spatial resolution [39].

### 2.2. Incident Report Analysis

Detection point information from fire reports was examined to help determine the structure of this sensitivity study. Integrated Reporting of Wildland-Fire Information

(IRWIN) [20] hosts incident report information for 1705 fires in the state of Colorado for the year 2018. Out of the 1705, 1292 fires (75.8%) show a shift in the detection point location within the fire event's incident report. A total of 27.9% of these fires had a detection point that shifted by less than 100 m, 25% had a detection point that shifted between 100 and 1000 m, and 47.1% had a detection point that shifted more than 1000 m. These shifts are representative of the uncertainty between the initial detection and the actual ignition location, and typically occur in the fire report days after the initial detection.

In addition to investigating the detection information in the incident reports, detection points were mapped alongside the first observed perimeter for fires with available data. This brief visual analysis shows that the detected location may be outside the first observed fire perimeters. Figure 1 shows an example using the Indian Valley Fire, which occurred in Colorado in 2018 and is used as a case study in this research. The detected point is approximately 135 m outside the closest location of the first observed burned area perimeter which was captured less than 5 h after the fire was detected. We include this information simply to highlight the uncertainty in the reported wildland fire detections.

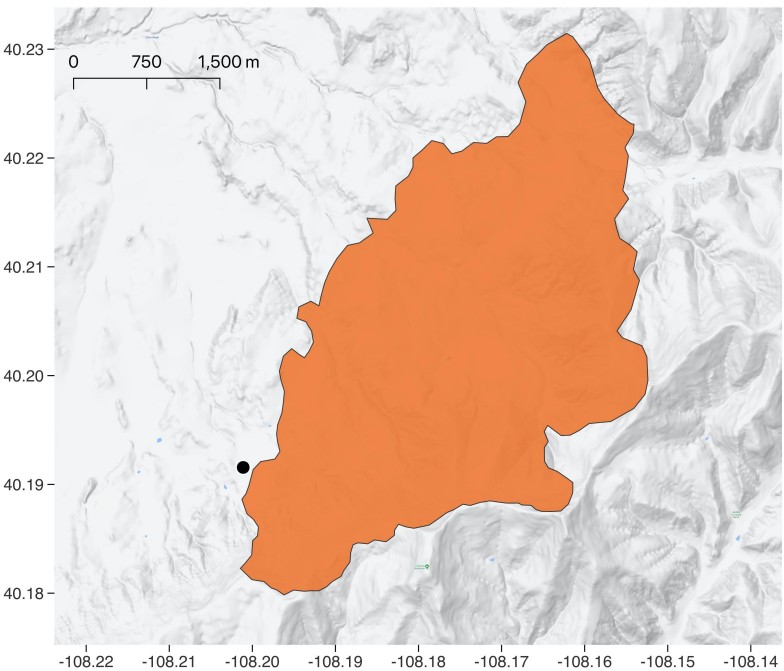

**Figure 1.** The first observed active fire perimeter (orange) for the Indian Valley Fire, Colorado 2018, and the fire's initial detection location (black dot). The initial detection location was obtained through the Integrated Reporting of Wildland-Fire Information system [20], and the first observed fire perimeter was obtained through the National Interagency Fire Center [40]. Longitude (decimal degrees) and latitude (decimal degrees) showing the location of this fire are given on the x-axis and y-axis, respectively.

### 2.3. Case Study Selection

Ten case studies of 2018 Colorado wildfires were selected from the National Fire Situational Awareness portal (National Interagency Fire Center) [40]. Out of the viable cases (cases with available detection data), an effort was made to choose case studies that spanned a variety of fuel and terrain characteristics, as well as a range of detection times. The search resulted in the cases mapped in Figure 2 and described in Table 1.

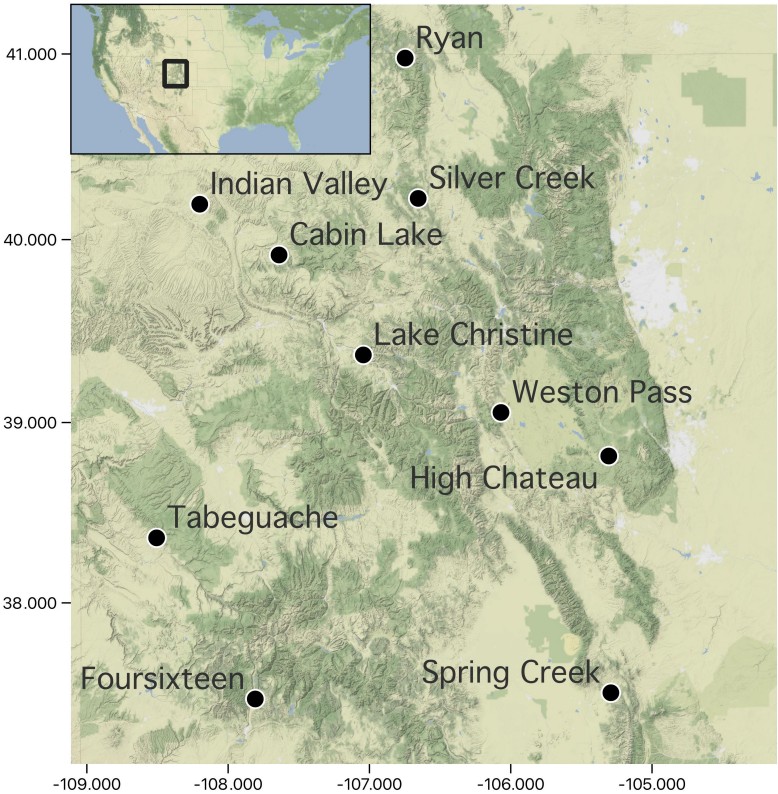

**Figure 2.** Ten Colorado wildland fire case studies from 2018, with initial detection locations (black dots). Initial detection locations for each case study were obtained from the Integrated Reporting of Wildland-Fire Information system [20]. Longitude (decimal degrees) and latitude (decimal degrees) showing the location of the fire case studies are given on the x-axis and y-axis, respectively.

**Table 1.** Case study fire names, detection dates, and detection times (UTC).

| Fire Name | Detection Date | Detection Time |
|-----------|----------------|----------------|
| 416 | 1 June 2018 | 1602 |
| Cabin Lake | 29 July 2018 | 2000 |
| High Chateau | 29 July 2018 | 2000 |
| Indian Valley | 20 July 2018 | 2030 |
| Lake Christine | 3 July 2018 | 0011 |
| Ryan | 16 September 2018 | 0136 |
| Silver Creek | 19 July 2018 | 2030 |
| Spring Creek | 27 June 2018 | 2130 |
| Tabeguache | 7 July 2018 | 0434 |
| Weston Pass | 28 June 2018 | 2030 |

For each of the case studies, a set of model initiation points was derived from the reported detection point. Eight points were selected in each of the cardinal and intercardinal directions 100 m away from the detection location, and likewise, eight more points were selected 1000 m away from the detection location as shown in Figure 3. The decision to calculate points 100 m and 1000 m away was determined through examination of the detection points recorded for 2018 Colorado wildfires, as described above. This covers the range of more than half of the shifts in detection location observed in the IRWIN record of 2018 Colorado fires.

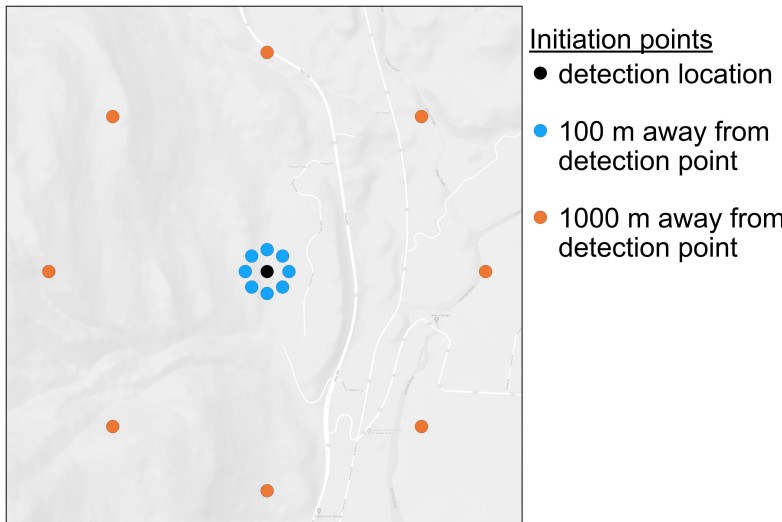

**Figure 3.** Configuration of derived initiation points, shown for the 416 Fire case study. The reported detection point is shown in black, derived points 100 m away are shown in blue, and 1000 m away are shown in orange.

In total, each case study has 17 initiation points for simulation; the detection point, along with the 16 derived points. WRF-Fire simulations were run for each of the case studies, at each of these 17 points starting from the time of detection. This isolated the changed variable in the model setup to the initiation point location, providing model output from each case study's set of simulations to determine the effects of shifting the initiation location on forecast burned area and propagation direction. Simulations were also run from each of the 17 points at plus and minus six hours from the detection time for each case study. For example, simulations of the 416 Fire were initiated at 1602 UTC (its detection time), as well as 1002 UTC and 2202 UTC. Simulating the case studies at these offset times provides data showing how the timing affects the modeled fire area and propagation direction.

Half of the case studies were detected in the early afternoon (14:00–15:30 MDT), with the remaining case studies detected either in the evening, late at night, or late morning. Starting simulations plus and minus six hours from these detection times provides a broad range of times to examine. The combined effects of shifts in both location and time are observed in the model output from simulations started at the shifted detection points and the augmented times. Cumulatively, each case study had 52 resulting forecasts to examine; one set of model output at each of the 17 locations and each of the three initiation times.

WRF-Fire provides model output at a user-defined time window. Model forecasts for this research are generated at an hourly time resolution. Specifically, the resulting model output from a six-hour lead time (forecast fire area and direction of propagation from the sixth simulated hour) was examined. A six-hour lead time, along with the total number of case studies, initiation time offsets, and initiation points, was chosen for two reasons; to keep within reasonable computational limits, and to generate output data with enough difference for comparison. For the sixth hour of each forecast, we calculated the forecast fire area as well as the direction of propagation. The direction of propagation is calculated using the direction of the vector from the initiation point to the centroid of the predicted fire area.

## 3. Results

Figures 4 and 5 summarize the overall results of this research. The range of simulated burned areas and propagation direction remains more compact for simulations initiated 100 m away from the detection point than for simulations initiated 1000 m away from the

detection point. However, no significant pattern is evident connecting the overall range of simulated fire areas and propagation directions to the time of day.

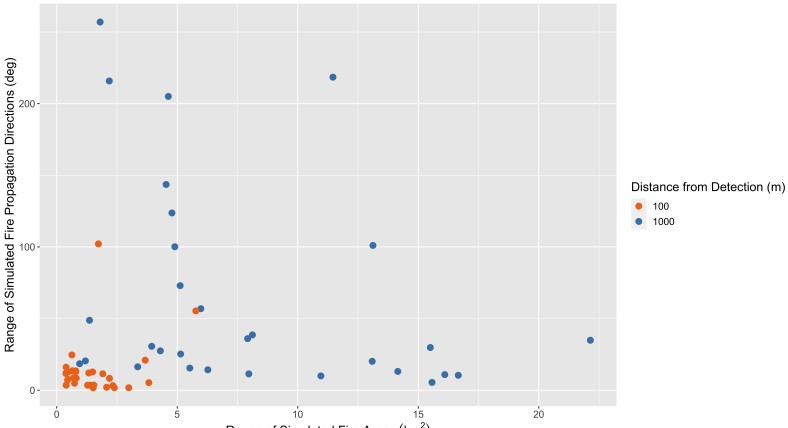

**Figure 4.** Distribution of range of values for the simulated burned area and propagation directions by distance from the detection point. Each point represents the values for a set of model results initiated from the points 100 m away from the detection point (orange) or 1000 m away from the detection point (blue) at the detection time, six hours before, or six hours after.

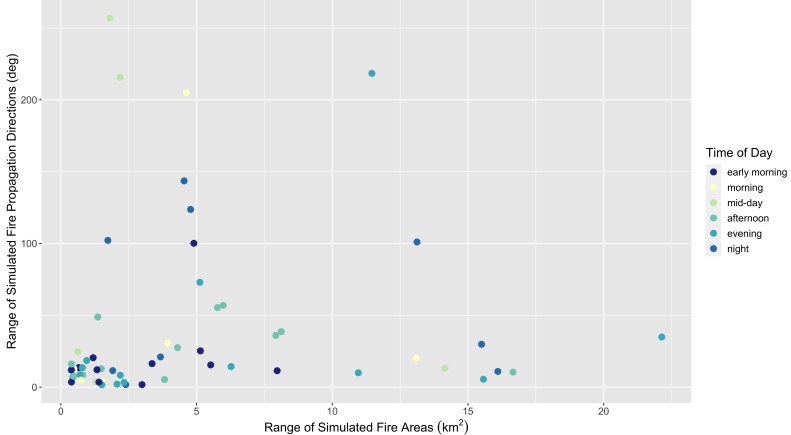

**Figure 5.** Distribution of the range of values for the simulated burned area and propagation directions by time of day. Each Time class represents four hours; "early": 2:00 a.m.–5:59 a.m., "morn": 6:00 a.m.–9:59 a.m., "mid": 10:00 a.m.–1:59 p.m., "aft": 2:00 p.m.–5:59 p.m., "eve": 6:00 p.m.–9:59 p.m., "night": 10:00 p.m.–1:59 a.m.

### 3.1. Effects of Initiation Point Location on Forecast Burn Area

Figure 6 summarizes the forecast fire area from cases initiated at their respective detection times. The range of fire area values has a tendency to broaden with distance away from the detected fire location. Across all ten case studies (initiated at their detection times), the average difference in predicted area was 1.29 km$^2$ for simulations started 100 m away from the detection location, and 8.49 km$^2$ for simulations started 1000 m away from their detection point. Simulations of the Silver Creek Fire had the smallest range of values, a difference between the smallest and largest simulated area of 0.72 km$^2$, while simulations of the Spring Creek Fire had the largest range, 15.56 km$^2$.

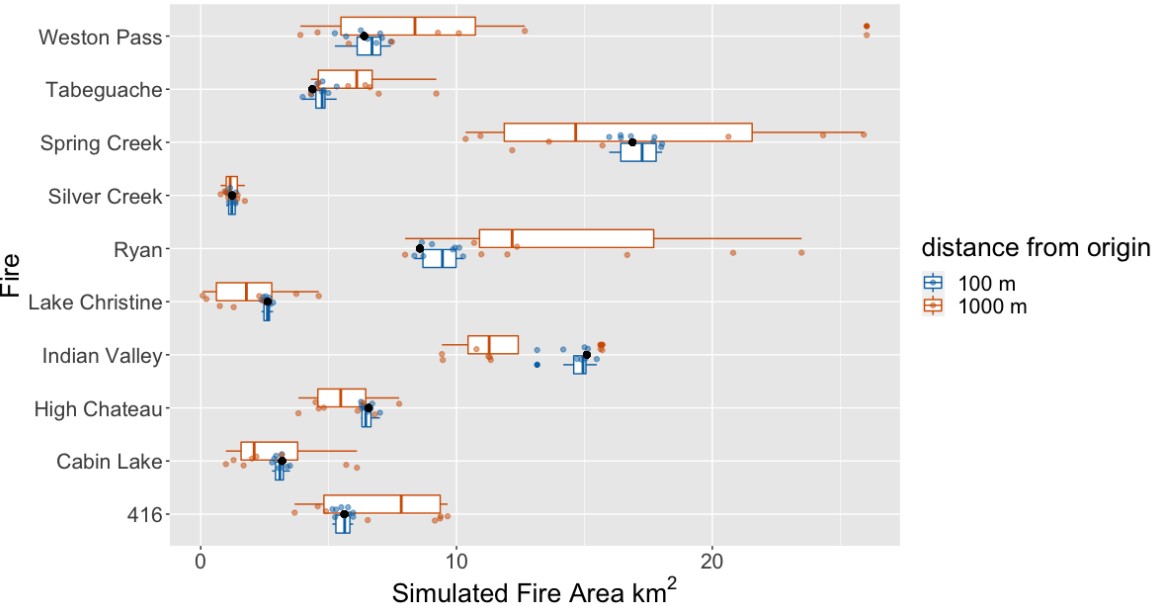

**Figure 6.** Forecast burned area for each case study resulting from simulations initiated at the detection time and all 17 initiation points.

### 3.2. Effects of Initiation Time on Forecast Area

The effect of shifting the simulation initiation time by six hours on the forecast area varies by case study. Considering simulations started from just the detected locations, the smallest range of forecast area values was (1.23 km$^2$–1.72 km$^2$) from the Silver Creek Fire case study, and the largest was (6.57 km$^2$–29.10 km$^2$) from the High Chateau Fire case study. Forecast fire area values simulated from the detected location at the detected time, plus and minus six hours for each case study are summarized in Figure 7 below.

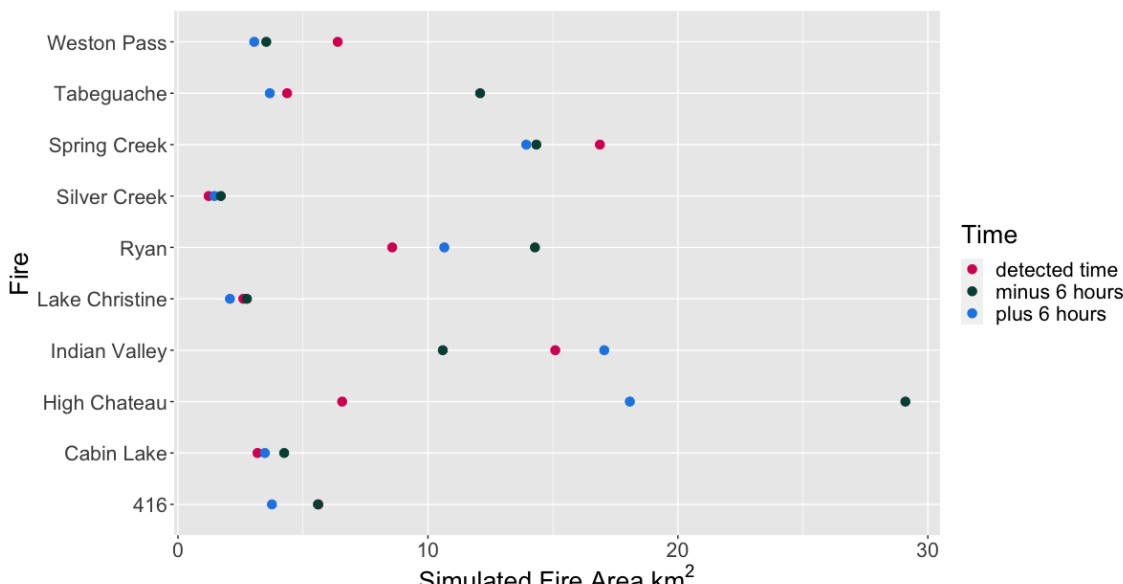

**Figure 7.** Forecast burned area at each case study's detection location initiated at the detection time (magenta), six hours before (dark green) and six hours after (blue).

### 3.3. Effects of Initiation Point Location on Forecast Propagation Direction

Figure 8 summarizes the forecast direction of propagation from case studies initiated at their respective detection times. Again, the range of values has a tendency to broaden with distance away from the detected fire location. Simulations of the Silver Creek Fire had

the smallest range of values (113.51°–132.00°), while simulations of the 416 Fire had the largest (317.48°–29.27°).

**Figure 8.** Forecast direction of propagation for case studies initiated at the time of detection. The range of propagation directions tends to be wider for simulations initiated 1000 m away (orange vectors) from the detection location than for those initiated 100 m away (blue vectors) from the detection location. he direction of the simulation initiated from the detection location is shown in black.

### 3.4. Effects of Initiation Time on Forecast Propagation Direction

Forecast propagation direction simulated from the detected location at the detected time, plus and minus six hours for each case study are summarized in Figure 9. Again, the effects vary by case study. Considering simulations started from just the detected locations, the smallest range of forecast fire propagation directions was (67.24°–72.34°) from the Weston Pass Fire case study, and the largest was (67.77°–257.54°) from the Cabin Lake Fire case study.

### 3.5. Effects of Both Initiation Point Location and Initiation Time on Forecast Area and Propagation Direction

Simulations of the 416 Fire are used for a closer look at the combined effects of initiation point location and time on simulated fire area and propagation direction. The results of this case study are summarized in Figure 10. The range of propagation directions is narrowest at the time of detection (10:02 a.m. MDT, 1602 UTC, a range of 71.79°), while the range of propagation directions is wider both six hours before (4:02 a.m. MDT, 1002 UTC, 213.70°) and six hours after the detection time (4:02 p.m. MDT, 2202 UTC, 162.91°). The range of simulated burned areas resulting from initiation locations 100 m away from the detected location are more compact than those resulting from initiation locations 1000 m away from the detected location. The area difference for simulations 100 m away from the detected location at the time of detection is 0.81 km$^2$, and 5.98 km$^2$ for simulations 1000 m away

from the detected location. The area differences for simulations initiated six hours before the detection time are 0.79 km$^2$ and 1.78 km$^2$ for initiation points 100 meters and 1000 m from the detection location, respectively. The area differences for simulations initiated six hours after the detection time are 1.73 km$^2$ and 4.78 km$^2$ for initiation points 100 m and 1000 m from the detection location, respectively.

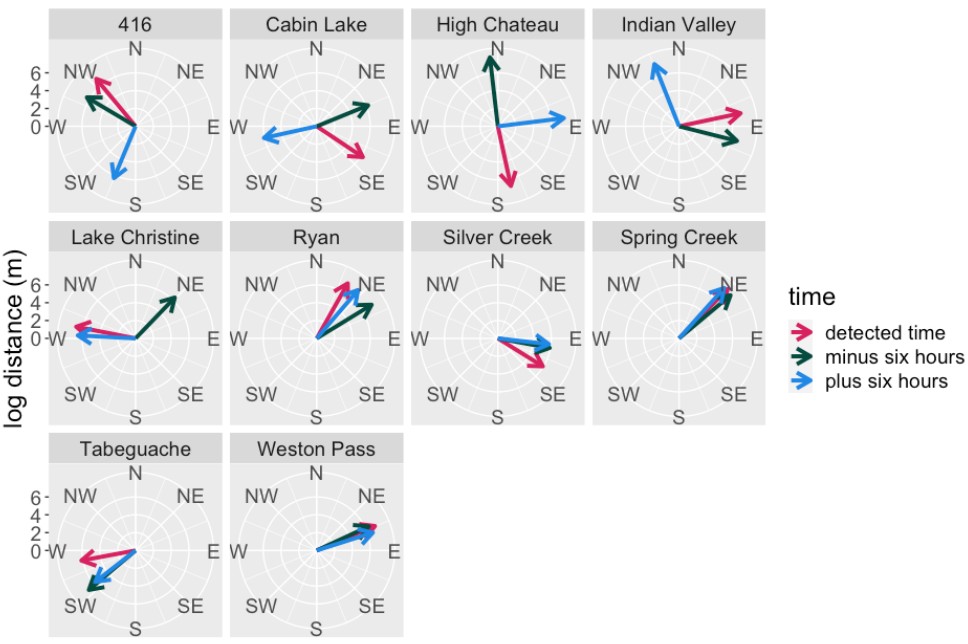

**Figure 9.** Forecast propagation direction for simulations initiated at the detection location at the time of detection (magenta), six hours before (dark green), and six hours after (blue).

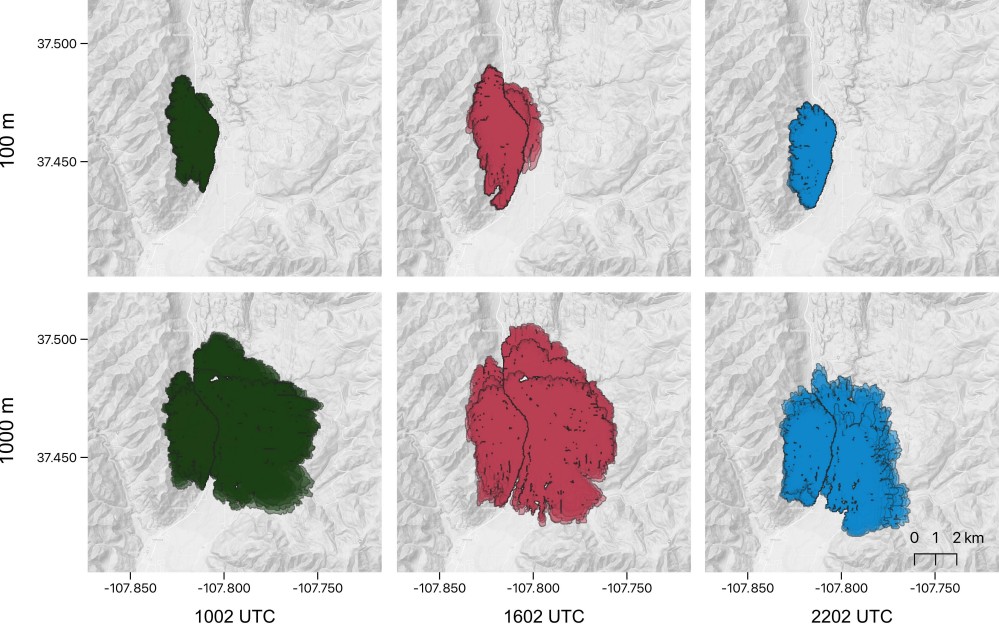

**Figure 10.** Forecast burned area perimeters for the 416 Fire case study. Simulations initiated at the detection location at the time of detection (magenta, center), six hours before (dark green, left), and six hours after (blue, right) at initiation points 100 m away from the detection points (top) and 1000 m away from the detection point (bottom). Longitude (decimal degrees) and latitude (decimal degrees) showing the location of the 416 Fire are given on the x-axis and y-axis, respectively.

## 4. Discussion

The model initiation location affects the range of forecast burned area and propagation direction values, while the time of day does not have the same significant influence over the model output. In particular, simulations initiated 1000 m away from the detection location had a broader range of forecast burn area values than simulations initiated 100 m away from the detection location, and the range of forecast areas differs by case study. The Rothermel rate of spread equations implemented in WRF-Fire rely on several input values. Among these, the initiation location and time determine the fuel model (a set of fuel characteristics including load depth and fuel type), the wind vectors, and the terrain characteristics. For each domain of the fire case studies used in this research, the fuel model, wind direction (at the 10 m height), wind speed (at the 10 m height), terrain aspect, and terrain slope were collected. The range of values for each of these variables was calculated for each simulation and used to fit a separate generalized linear model for each output variable (forecast burned area and propagation direction). Generalized linear models are used here to estimate the relationship between variables. Summaries for the models are shown in Tables 2 and 3. The range of fuel models, wind directions and speeds, and aspects are important predictors of the range of forecast burned area values. The same is true of the range of forecast propagation directions, though the range of slope values is also important in this case. Essentially, this brief investigation shows that the range of forecast burned area and propagation direction values is related to the heterogeneity of the fire domain. In a space with consistent fuel models, slopes and aspects, with little variability in the wind speeds and wind directions across space and time, forecasts initiated at different locations and different times will result in similar burned areas and propagation directions. In a space with several different fuel models and more complex terrain, with variability in the wind speeds and wind directions across space and time, forecasts initiated at different locations and different times will result in wider ranges of burned area and propagation direction values.

The case studies selected for this research are located in the state of Colorado, situated in the Western United States. Colorado is an arid state with a wide range of ecosystems including alpine tundra and short-grass prairie. Effort was made to ensure that the case studies encompassed a variety of meteorological, terrain, and fuel conditions by examining the ranges of wind speed, wind direction, fuel model, slope, and aspect values for each case study during the selection process. While the conditions in the state of Colorado are not representative of all environments, the results of this work should translate well to similar environments. In particular, we would expect analogous results in wooded foothills, rugged mountains with coniferous forests, and rolling grassland prairies. In less arid regions, fuel moisture may have a dampening effect on the simulated fire spread rate, in which case the ignition location and time of day may have less influence on the forecast burned area and propagation direction. Additional studies are needed to determine the sensitivity of WRF-Fire output to ignition location and time in other environments.

**Table 2.** Generalized Linear Model Summary for Modeling the Range of Forecast Burn Area Values (area_range) from the Range of Fuel Models (n_fuel), Terrain Slope (slope_range), Aspect (aspect_range), 10 m Height Wind Direction (wdir_range), and 10 m Height Wind Speed (wspd_range). Each variable is shown with its respective regression coefficient, statistical significance (asterisks), and root mean squared error (RMSE, in parentheses).

|  | Dependent Variable |
| --- | --- |
|  | Area_range |
| n_fuel | −0.351 *** |
|  | (0.079) |
| slope_range | 0.001 |
|  | (0.009) |
| aspect_range | 0.005 *** |
|  | (0.001) |
| wdir_range | 0.010 *** |
|  | (0.004) |
| wspd_range | 0.119 * |
|  | (0.065) |
| Constant | 1.530 *** |
|  | (0.267) |

Note: * $p < 0.1$; ** $p < 0.05$; *** $p < 0.01$.

**Table 3.** Generalized Linear Model Summary for Modeling the Range of Forecast Propagation Direction Values (dir_range) from the Range of Fuel Models (n_fuel), Terrain Slope (slope_range), Aspect (aspect_range), 10 m Height Wind Direction (wdir_range), and 10 m Height Wind Speed (wspd_range). Each variable is shown with its respective regression coefficient, statistical significance (asterisks), and root mean squared error (RMSE, in parentheses).

|  | Dependent Variable |
| --- | --- |
|  | Dir_range |
| n_fuel | 0.112 *** |
|  | (0.027) |
| slope_range | 0.057 *** |
|  | (0.003) |
| aspect_range | 0.002 *** |
|  | (0.0003) |
| wdir_range | 0.017 *** |
|  | (0.001) |
| wspd_range | 0.085 *** |
|  | (0.023) |
| Constant | 1.395 *** |
|  | (0.126) |

Note: * $p < 0.1$; ** $p < 0.05$; *** $p < 0.01$.

While there are other possible sources of uncertainty in wildland fire behavior model inputs, this study shows the importance of using the best available ignition time and location data. For implementing wildland fire behavior models in the context of operational fire management, the detection information and any in-situ observations may be the best available at the time. In the context of wildland fire research, bounding the ignition information using remotely sensed data, as done in the work by Benali et al. (2017) [18] will aid in improving the model setup. The ability to do so will likely improve in the near future as higher-frequency, higher-resolution remotely sensed data becomes available. This advancement in technology will certainly inform wildland fire research, and has the potential to help provide near real-time active fire data, including ignition information.

## 5. Conclusions

As shown, the range of forecast burned area and propagation direction values is strongly affected by the location of the initiation location and to a lesser degree by the time of initiation. This is due to the input data associated with the initiation location and time. The location determines the fuel model and terrain characteristics, while both the time and location determine the wind conditions. While the case studies presented in this research are located in the state of Colorado in the United States, the results should hold in other environments, as care was taken to select case studies with a variety of meteorology, terrain, and fuel characteristics. Incident commanders and wildland fire researchers using wildland fire behavior models can anticipate that the range of forecast burned areas and propagation directions will be wider in heterogeneous fire domains, and more compact in homogeneous spaces.

**Author Contributions:** Conceptualization, A.D., A.S.-A., J.C.K. and B.G.B.; methodology, A.D.; software, A.D.; validation, A.D., E.S. and A.S.-A.; formal analysis, A.D.; investigation, A.D. and E.S.; resources, A.D.; data curation, A.D.; writing—original draft preparation, A.D.; writing—review and editing, A.D., A.S.-A., J.C.K., B.K., B.G.B.; visualization, A.D.; supervision, A.S.-A., B.G.B., J.C.K., B.K. and J.K.B.; project administration, J.C.K.; funding acquisition, B.G.B. and J.C.K. All authors have read and agreed to the published version of the manuscript.

**Funding:** Funding was provided by the State of Colorado through the Center of Excellence for Advanced Technology Aerial Firefighting, Division of Fire Prevention and Control through contract number 85593. We would also like to acknowledge high-performance computing support from Cheyenne (doi:10.5065/D6RX99HX) provided by NCAR's Computational and Information Systems Laboratory, sponsored by the National Science Foundation.

**Institutional Review Board Statement:** Not applicable.

**Informed Consent Statement:** Not applicable.

**Data Availability Statement:** WRF-Fire may be found at the GitHub page: https://github.com/openwfm/wrf-fire; Case study data may be found at the National Interagency Fire Center website: https://www.nifc.gov and the Integrated Reporting of Wildland-Fire Information mapping interface: https://nifc.maps.arcgis.com/apps (accessed on 10 February 2022).

**Acknowledgments:** Thank you to Clara Chew and Michael Koontz for your thoughtful revisions.

**Conflicts of Interest:** The authors declare no conflict of interest.

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
