# Peer review of "Weather Research and Forecasting—Fire Simulated Burned Area and Propagation Direction Sensitivity to Initiation Point Location and Time"

_fire, doi:10.3390/fire5030058_

Round 1

Reviewer 1 Report

Review on Ms. Ref. No.: fire-1648299

WRF-Fire Simulated Burned Area and Propagation Direction Sensitivity to Initiation Point Location and Time

Amy DeCastro, Amanda Siems-Anderson, Ebone Smith, Jason C. Knievel, Branko Kosovi´c, Barbara G. Brown, and Jennifer K. Balch

Recommendation: Accept after minor revision

  1. Summary

The authors of the submitted manuscript examine the range of burned area values and propagation directions resulting from different initiation point locations and time based on the Colorado 2018 wildland with a total of 520 case study simulations. The methodology is very good, strict, synthesized, and appropriate, showing the big capacity of the software. I agree with the main conclusion that the range of forecast burned area and propagation direction values is strongly affected by the location of the initiation location, as clearly shown by the research. Well done research!

  1. Major issues
  • The literature review part in the Introduction shall be improved. For example, it is a pity for your paper, not to step on the wildland fire research of ADAI_CEIF Forest Fire Research Center in Portugal (www.adai.pt).
  1. Minor issues
  • Avoid using abbreviations in the title, please. Although this is the commonly used name of this software, abbreviations in the title more distract the reader’s attention than attract him.
  • The authors need to add the country in their address information. Fire is a big international journal, not a local American one.
  • Line 71: What does this “?” mean?
  • References for the Figure 1 and Figure 2 maps should appear in the figure labels.
  1. Opinion

Despite my remarks above, I assess the overall manuscript as very good and I recommend it for acceptance after minor revision.

Kind regards!

Author Response

We thank the reviewer for their thoughtful suggestions. Please see the attached document addressing each comment. 

Much appreciated, 

Amy DeCastro

Reviewer 2 Report

The paper studies how the propagation direction of burned areas are more affected by the location of the initial ignition than the time of ignition. The study is well presented with strong simulation results.

Main Revisions:

There is a Missing Reference in line 70/71: "the fire-behavior physics module that implements the Rothermel rate of spread equations 70
[? ]"

Table 2 and Table 3 should be revised in order to understand the values. Are the values in paranthesis the standard deviation? Is it the p-value? If it is the p-value then the notes don’t seem to be matching. Please make it more clear.

In the Conclusions Section it would be interesting to theorize if the results hold up in different regions, with more vegetation or less vegetation. Could time be of ignition affect more the burned area in areas with different landscapes or areas where the detection of fires could be longer than the simulated 6 hours. It would also be of good importance to make suggestion for future works.

Author Response

(The authors gave the same response as above.)

Reviewer 3 Report

The reviewed article concerns the modeling of forest fire development with the use of such a tool as WRF-Fire. The authors focused on determining the effect of the accuracy of determining the location of the ignition point and the time of fire initiation on the burnt surface and the direction of the fire spread. In my opinion, the literature review on this subject presented in Chapter 1 is too modest and does not take into account all relevant items. The consequence of this is too few literature sources listed in the references. Moreover, the quantitative method used was described too generally. For example, there is no overview of the basic algorithms used during the simulation. Moreover, in my opinion, the analysis of the results could be more detailed and, as it stands, it covers some conclusions that should be moved to the last chapter. This way, some repetitions could be avoided. In addition to these general observations, some specific notes are provided below:

Line 71 – instead of the source number, a question mark is shown in square brackets,

Figures 1,2 and 10 - no description of the horizontal and vertical axis,

Line 131 - MDT time should be written in the following format hour: minute,

Line 140 - the word "the" before "this research" should be removed,

Figures 4 and 5 - no units on the coordinate axes,

Figure 4 - one word "the" before "the values" should be removed in the title of the figure,

Figure 5 - full names should be used instead of abbreviations in the legend "Time", e.g. instead of "morn", "morning" etc. Similar in the title of the drawing,

Figures 6 and 7 - The last sentence in the title of both pictures should be moved to chapter 4 titled "Discussion",

Line 208 - why the assumed wind direction and speed were not given, only the height of 10 m. This should be explained in more detail,

Tables 2 and 3 - why there are two values in the tables, one without parentheses and the other in parentheses. This should be explained in more detail. Where did these values come from? There is no explanation of the size of p,

Lines 236-238 - this sentence was repeated from Chapter 4,

Line 250 - If there was no financing, this point should be deleted.

Author Response

(The authors gave the same response as above.)

Round 2

Reviewer 3 Report

I would like to thank the authors for a solid improvement of the article in accordance with my comments and a detailed description of the changes in the content of the manusript